# Adjuvant Curdlan Contributes to Immunization against *Cryptococcus gattii* Infection in a Mouse Strain-Specific Manner

**DOI:** 10.3390/vaccines10040620

**Published:** 2022-04-15

**Authors:** Patrícia Kellen Martins Oliveira-Brito, Gabriela Yamazaki de Campos, Júlia Garcia Guimarães, Letícia Serafim da Costa, Edanielle Silva de Moura, Javier Emílio Lazo-Chica, Maria Cristina Roque-Barreira, Thiago Aparecido da Silva

**Affiliations:** 1Department of Cell and Molecular Biology and Pathogenic Bioagents, Ribeirão Preto Medical School, University of São Paulo, Ribeirão Preto 14049-900, SP, Brazil; patriciakellen@usp.br (P.K.M.O.-B.); gabyamazaki@usp.br (G.Y.d.C.); jgguimaraes@usp.br (J.G.G.); mouraedanielle@gmail.com (E.S.d.M.); mcrbarre@fmrp.usp.br (M.C.R.-B.); 2Department of Microbiology, Institute of Biomedical Sciences, University of São Paulo, São Paulo 14049-900, SP, Brazil; leticiaserafim@usp.br; 3Institute of Natural and Biological Sciences, Federal University of Triângulo Mineiro, Uberaba 38025-189, MG, Brazil; javier.chica@uftm.edu.br; 4Thiago Aparecido da Silva, Departamento de Biologia Celular e Molecular e Bioagentes Patogênicos, Faculdade de Medicina de Ribeirão Preto, Universidade de São Paulo, Avenida Bandeirantes 3900, Ribeirão Preto 14049-900, SP, Brazil

**Keywords:** *Cryptococcus gattii*, immunotherapy, curdlan, β-glucan peptide, Dectin-1

## Abstract

The low efficacy and side effects associated with antifungal agents have highlighted the importance of developing immunotherapeutic approaches to treat *Cryptococcus gattii* infection. We developed an immunization strategy that uses selective Dectin-1 agonist as an adjuvant. BALB/c or C57BL/6 mice received curdlan or β-glucan peptide (BGP) before immunization with heat-killed *C. gattii*, and the mice were infected with viable *C. gattii* on day 14 post immunization and euthanized 14 days after infection. Adjuvant curdlan restored pulmonary tumor necrosis factor- α (TNF-α) levels, as induced by immunization with heat-killed *C. gattii.* The average area and relative frequency of *C. gattii* titan cells in the lungs of curdlan-treated BALB/c mice were reduced. However, this did not reduce the pulmonary fungal burden or decrease the i0,nflammatory infiltrate in the pulmonary parenchyma of BALB/c mice. Conversely, adjuvant curdlan induced high levels of interferon-γ (IFN-γ) and interleukin (IL)-10 and decreased the *C. gattii* burden in the lungs of C57BL/6 mice, which was not replicated in β-glucan peptide-treated mice. The adjuvant curdlan favors the control of *C. gattii* infection depending on the immune response profile of the mouse strain. This study will have implications for developing new immunotherapeutic approaches to treat *C. gattii* infection.

## 1. Introduction

Approximately 1.6 million people each year have been estimated to die from fungal infections worldwide [1]; however, invasive fungal infections remain neglected by public health policies [1,2]. According to the Global Action Fund for Fungal Infections, the genus *Cryptococcus* has been responsible for clinical cases of cryptococcosis, which has highly impacted global public health over the last several decades [3,4]. The incidence of cryptococcosis in immunosuppressed or healthy individuals has been attributed to the presence of two species of *Cryptococcus*. *C. neoformans* commonly affects immunocompromised patients, such as individuals who have undergone transplantation and are therefore being treated with corticosteroids, antineoplastic therapies, and other immunosuppressive drugs. However, a majority of the *C. neoformans* cases arise in patients with acquired immune deficiency syndrome (AIDS) [5]. Conversely, *Cryptococcus gattii* is considered a primary pathogen because of its ability to infect healthy individuals. *C. gattii* has a predilection for the pulmonary tissue [6,7], whereas *C. neoformans* is initially found in the lungs and subsequently spreads to the central nervous system [8].

Cryptococcosis occurs via inhalation of propagules sourced from *C. neoformans* or *C. gattii*, which then enter and lodge in the pulmonary alveoli [9]. Resident alveolar macrophages, neutrophils, and dendritic cells (DCs) can recognize *Cryptococcus* spp. and initiate an immune response against the fungus [10,11]. Earlier studies on *C. neoformans* infection suggest that *Cryptococcus* spp. is recognized by innate immune cells, such as macrophages and DCs, via pattern recognition receptors (PRRs) [10]. Dectin-1 and toll-like receptors (TLRs) are PRRs localized on the cell surface and recognize the highly conserved pathogen-associated molecular patterns (PAMPs) on *Cryptococcus* spp. [12,13,14]. Dectin-1 and TLRs act as sensors during infection and play crucial roles in maintaining the balance between clearance and fungal dissemination. The β-glucan in the inner layer of the *Cryptococcus* spp. capsule can be recognized by Dectin-1. However, this recognition is inhibited by the outer layer of the capsule, which is enriched in polysaccharides that mask the β-glucan. Cryptococcal polysaccharides are mainly composed of two major virulence factors that have important immunomodulatory properties [15]: glucuronoxylomannan (GXM), which can directly interfere with T lymphocyte function [16], induce FasL expression in macrophages causing death in nearby T cells [17,18], and affect macrophage cell proliferation and trigger apoptosis [18,19]; and galactoxylomannan (GalXM), which can induce dysregulation of cytokines and apoptosis of T lymphocytes [20], in addition to negatively regulating the T cell response [18,21,22]. Furthermore, GXM, the major capsule polysaccharide of *Cryptococcus* spp., can be recognized by TLR2 and/or TLR4 [10,23]. However, Dectin-1, TLR2, and TLR4 are not required for host defense against *C. neoformans* or *C. gattii* [13,24,25,26,27], and GXM strongly regulates the host immune response [15,23,28,29]. Thus, the capacity of innate immune cells to recognize *Cryptococcus* spp. via PRRs is often compromised by the components of the capsule.

The PRRs on innate immune cells recognize pathogen-derived PAMPs, which enables the induction of the host immune response that controls the infection and modulates the activation of the adaptive immune response. T-cell activation requires three signals: a ligand that is bound to the major histocompatibility complex class II (MHC-II) molecules on the surface of an antigen-presenting cell (APC); a second signal mediated by costimulatory molecules, such as CD80/CD86; and a third signal received in the form of cytokines that induce the T cell differentiation. This immunological synapse drives the activation, differentiation, and migration of T helper (Th) cells and thereby enables the development of a pathogen-specific immune response. The precise activation of innate immune cells is important for inducing Th cell responses, which can thereafter mediate the production of antibodies against cryptococcosis and make the relevant changes [30,31,32]. Differentiation of Th1 and Th17 cells is necessary for pulmonary clearance and protection against *C. neoformans*, as verified in animal models [31,33,34]. However, *C. gattii* affects the differentiation of Th1 and Th17 cells through the functional attenuation of DCs and the reduction of chemokine expression in the lungs [35]. *C. gattii* infection blocks the release of tumor necrosis factor-α (TNF-α) and chemokines, subsequently impairing the maturation of pulmonary DCs [35,36]. Furthermore, this fungus promotes immunoparalysis of DCs via the retention of phagosomal F-actin [37]. The *C. gattii* strains R265 and R272 employ these immune evasion mechanisms and compromise the induction of adaptive immune responses [36,37]. Therefore, it is necessary to develop immunotherapeutic strategies involving cryptococcal components, vaccine delivery systems, and adjuvants from distinct sources [31,38,39,40,41]. 

The administration of immunomodulatory molecules has demonstrated promising effects for the treatment of fungal infections, such as *Candida albicans* and *Paracoccidioides brasiliensis* infections, which were controlled by the immunomodulatory activities of ArtinM [42,43] and Paracoccin [44,45] lectins, and the P10 peptide [46]. These immunomodulators were prophylactically administrated one or three days before infection and caused a reduction in the fungal burden in the lungs [42,43,44,45,46]. Carbohydrate-based adjuvants, such as β-glucans, are known to stimulate and regulate the immune response while exhibiting high biocompatibility and low toxicity [47,48]. Curdlan is a linear β-glucan derived from *Alcaligenes faecalis*, and β-glucan peptide (BGP) is a polysaccharide with a highly-branched glucan portion derived from *Trametes versicolor*. Both curdlan and BGP are specifically recognized by the Dectin-1 receptor on macrophages and DCs [49]. The binding of curdlan to Dectin-1 initiates a signal cascade via the Syk kinase. It leads to the production of reactive oxygen species, mitogen-activated protein kinase (MAPK) activation, nuclear factor kappa-light-chain-enhancer of activated B cells (NF-κB) phosphorylation, and subsequent cytokine production [50,51]. These events promote the maturation and migration of macrophages and DCs that drive the differentiation of Th1 and Th17 cells [40,50,52,53,54,55,56,57,58]. Previous studies have demonstrated that curdlan induces a strong Th17 response [59,60]. Consequently, the potential of curdlan to activate Th1 and Th17 responses over time during *C. gattii* infection must be evaluated. The ability of curdlan or BGP to serve as an immunotherapeutic agent against *C. gattii* infection has not been well studied. Here, we examined the adjuvancity of curdlan or BGP, when used in combination with heat-killed *C. gattii*. 

In this study, BALB/c mice immunized with heat-killed *C. gattii* showed increased CD11b^+^, Ly6G^+^, and CD11c^+^ cells in the lungs. However, the addition of adjuvant curdlan to the immunization protocol produced a different response, resulting in high levels of TNF-α and reduced relative expression of *Arginase-1* in the lungs. Moreover, immunizing the mice with heat-killed *C. gattii* combined with the adjuvant curdlan did not control the pulmonary burden of *C. gattii* in BALB/c mice. Morphometric analysis of the lungs from mice immunized with heat-killed *C. gattii* associated with the adjuvant curdlan revealed a significant increase in the number of yeast cells per pulmonary area. However, there was a reduction in the average area of *C. gattii* yeast cells, which was consistent with a decrease in the frequency of titan cells (yeast with diameter > 10 µm). In addition, immunizing C57BL/6 mice, which are more susceptible to infection with *Cryptococcus* spp. [61,62,63,64], with heat-killed *C. gattii* and curdlan, restored the interleukin (IL)-10 and IL-4 levels and increased the relative expression of *ROR-γt* compared to immunized mice alone. The administration of curdlan contributed to the reduction in the *C. gattii* burden in the lungs of C57BL/6 mice.

## 2. Materials and Methods

### 2.1. Animals

Male BALB/c and C57BL/6 mice (6–8 weeks old) were acquired from the animal house at the Campus of Ribeirão Preto, University of São Paulo, Ribeirão Preto, São Paulo, Brazil. They were maintained under standard, optimized hygienic conditions in the animal house of the Department of Cellular and Molecular Biology of the Ribeirão Preto Medical School, University of São Paulo. All experiments were approved by the Committee of Ethics in Animal Research of the College of Medicine of Ribeirão Preto at the University of São Paulo (protocol no. 192/2018). Five mice were used per group, and all experiments were independently performed twice.

### 2.2. Curdlan and BGP as Adjuvant

Curdlan, a β-1,3-glucan from *Alcaligenes faecalis* and a Dectin-1 agonist (Cat# tlrl-curd; InvivoGen, San Diego, CA, USA), was resuspended in phosphate-buffered saline (PBS; 0.1 M) with sodium hydroxide. BGP from *Trametes versicolor* is a Dectin-1 ligand (Cat # tlrl-bgp; InvivoGen, San Diego, CA, USA) and was resuspended in water.

### 2.3. Administration of a Dectin-1 Ligand before Immunization with Heat-Killed C. gattii

The *C. gattii* strain R265 was cultivated on Sabouraud dextrose (SD) (Cat# K25-1205, Kasvi, Brazil) agar at 30 °C with continuous agitation at 180 rpm. To obtain *C. gattii* yeast cells with a thin capsule (*C. gattii*-thin), an inoculum was prepared in yeast extract–peptone–dextrose (YPD) medium containing 0.5 M NaCl, as described by Ikeda-Dantsuji et al. [65], after incubation at 30 °C and agitation at 180 rpm for 20 to 24 h. The *C. gattii*-thin was heat-inactivated at 70 °C for 1 h (HK-*C. gattii*-thin). The cell concentration was determined using a Neubauer chamber with China ink. A fraction of the HK-*C. gattii*-thin yeast was spread on SD agar to confirm the heat inactivation.

BALB/c or C57BL/6 mice received 100 µL of curdlan (200 µg/mouse), BGP (200 µg/mouse in 200 µL), or PBS via the intraperitoneal route (i.p.) on day zero. After 3 days, the mice were anesthetized with ketamine hydrochloride (20.0 mg/kg, i.p.) and xylazine hydrochloride (2 mg/kg, i.p.). After confirming the absence of reflexes, the mice were immunized with 2 × 10^7^ HK-*C. gattii*-thin via the intranasal route (i.n.). HK-*C. gattii*-thin was administered twice, with an interval of 14 days between each administration (Figure 1A). On day 45 (14 days post immunization [d.p.i.]), the mice were anesthetized, and 1 × 10^5^ viable *C. gattii* yeast cells were inoculated (i.n.). On day 59 (14 days post infection), mice were euthanized. All assays were performed independently in triplicates, and five mice were used for each group.

### 2.4. Measurement of Serum IgM and IgG anti-GXM Titers

Serum IgM and IgG anti-GXM titers were measured by enzyme-linked immunosorbent assay (ELISA) on days 38, 45, and 59, as described previously [66,67]. The 96-well microtiter plates were coated with 20 µg/mL GXM, which was isolated from *C. gattii* as described by Wozniak and Levitz [68], and incubated overnight at 4 °C. The plate was washed three times with wash buffer (PBS-Tween 20; 0.05%) and blocked with MOLICO^®^ skim dry milk powder (1.0%) for 2 h at 25 °C (room temperature [RT]). The plates were washed five times, and mouse serum was added at serial dilutions of 1:10, 1:20, 1:40, 1:80, 1:160, 1: 320, and 1:640 for measuring anti-GXM IgM and 1:50, 1:100, 1:200, 1:400, 1:800, 1:1600, and 1:3200 for measuring anti-GXM IgG, as specified in Figure 1D and Figure 9. The serum was incubated for 2 h at 25 °C. The plate was washed five times before the addition of goat anti-mouse IgM (μ-chain specific) peroxidase antibody (1:2500, Cat# A8786, Sigma-Aldrich, St. Louis, MO, USA) or rabbit anti-mouse IgG (whole molecule) peroxidase antibody (1:5000, Cat# A9044, Sigma-Aldrich). After 1 h of incubation, the plates were washed seven times, and 50 µL of 3,3′,5,5′-tetramethylbenzidine (TMB) (Cat# T2885, Lot# BCBD9116V—Sigma-Aldrich, Inc., St. Louis, MO, USA) substrate solution was added. After 30 min of incubation at RT, 30 µL of stop solution (2N H_2_SO_4_) was added. Absorbance was read at 450 nm using a spectrophotometer (Power Wave X; BioTek Instruments, Winooski, VT, USA).

### 2.5. Isotyping Serum Immunoglobulins

Mouse serum was used to quantify the levels of immunoglobulin G1 (IgG1), IgG2a, IgG2b, IgG3, IgM, and IgA, and the kappa and lambda light chains using the Easy-Titer Mouse IgG Assay Kit (Cat# 88-50630-88, Thermo Fisher, Waltham, MA, USA), according to the manufacturer’s instructions.

### 2.6. Quantification of Pulmonary Fungal Burden

On day 59 of the immunization protocol (14 days post infection with *C. gattii*), mice lungs were aseptically removed and homogenized in a tissue homogenizer (IKA^®^; Werke, Staufen, Germany) using flasks containing sterile 1× PBS (pH 7.2). Next, 50 µL of the homogenate from each mouse was spread in duplicates on SD agar (Oxoid, Basingstoke, England, UK) for performing the colony-forming unit (CFU) assay. The plates were incubated at 30 °C for up to 48 h, and the number of colonies of *C. gattii* was normalized per gram of organ (CFU/mg).

### 2.7. Measurement of Pro- and Anti-Inflammatory Cytokines in the Lungs

Cytokine levels in the lung homogenates were measured by ELISA using the OptEIA™ Kit (BD Pharmingen, San Diego, CA, USA), according to the manufacturer’s instructions. TNF-α, IFN-γ, IL-12p40, IL-10, IL-6, IL-4, and IL-17A were quantified, the values were expressed in pg/mL using Prism 9.0 (GraphPad Software, San Diego, CA, USA), and a standard curve was plotted for each cytokine. The cytokine values were expressed as fold-changes, determined by the ratio between each sample and the mean values of the untreated group.

### 2.8. Phenotyping of Innate and Adaptive Immune Cells in the Lungs Using Flow Cytometry

Pulmonary leukocytes were obtained as previously described by Oliveira-Brito et al. [69]. Prior to phenotyping the cell populations in the lungs, the concentrations of the cell suspensions were determined using a Neubauer chamber. Each sample was adjusted to a concentration of 1 × 10^6^ cells/mL, and the cells were thereafter stained with fluorophore-conjugated antibodies. The following antibodies were obtained from BD Pharmingen (San Diego, CA, USA): anti-CD3 (PE-Cy5 rat anti-mouse CD3, clone 17A2), anti-CD11c (Alexa Fluor 488 rat anti-mouse CD11c, clone M1/70), anti-Ly6G (PE rat anti-mouse Ly-6G, clone 1A8), and anti-CD11b (PE rat anti-mouse CD11b, clone M1/70). After incubation for 30 min at 4 °C with 0.5 μg of anti-CD16/CD32 mAb (Fc block, clone 2.4G2, BD Pharmingen), 2.5 μg/mL of the aforementioned antibodies was added and the suspension incubated for 45 min at 4 °C. The cells were washed twice with PBS and fixed with 1% PBS-formaldehyde for further analysis using flow cytometry (Guava EasyCyte™ Mini System).

### 2.9. Measurement of Relative Gene Expression of Innate and Adaptive Immune Cell Differentiation Markers

Total RNA was isolated from lung cells using TRIzol^®^ reagent (Invitrogen™—Life Technologies Corporation, Carlsbad, CA, USA) according to the manufacturer’s instructions. Reverse transcription of RNA into complementary DNA (cDNA) was performed using an iScript cDNA Synthesis Kit (Bio-Rad Laboratories, Inc., Hercules, CA, USA). Real-time quantitative reverse transcription polymerase chain reaction (qRT-PCR) was performed in a 10 μL reaction with SsoFast™ EvaGreen^®^ Supermix (Bio-Rad Laboratories, Inc.). The reactions were read in the CFX96 Touch Real-Time PCR Detection System (Bio-Rad Laboratories, Inc.) and performed under the following conditions: 95 °C for 30 s, followed by 30 cycles of 95 °C for 5 s and 60 °C for 5 s, and melt curve at 65–95 °C for 5 s. Gene expression was quantified using the 2^−ΔΔCt^ method and normalized to β-actin. The PCR primers used were as follows: *β-actin* (F: 5′-AGCTGCGTTTTACACCCTTT-3′/R: 5′-AAGCCATGCCAATGTTGTCT-3′), *T-bet* (F: 5′-CACTAAGCAAGGACGGCGAA-3′/R: 5′-CCACCAAGACCACATCCAC-3′), *GATA-3* (F: 5′-AAGAAAGGCATGAAGGACGC-3′/R: 5′-GTGTGCCCATTTGGACATCA-3′), *ROR-γt* (F: 5′-TGGAAGATGTGGACTTCGTT-3′/R: 5′-TGGTTCCCCAAGTTCAGGAT-3′), *FOXP3* (F: 5′-ATTGAGGGTGGGTGTCAGGA-3′/R: 5′-TCCAAGTCTCGTCTGAAGGCA-3′), *iNOS2* (F: 5′-CCGAAGCAAACATCACATTCA-3′/R: 5′-GGTCTAAAGGCTCCGGGCT-3′), and *Arginase-1* (F: 5′-GTTCCCAGATGTACCAGGATTC-3′/R: 5′-CGATGTCTTTGGCAGATATGC-3′).

### 2.10. Histopathological and Morphometric Analysis of the Pulmonary Tissue and Its C. gattii Burden

For histopathological and morphometric analyses, the lungs were perfused with 5 mL of PBS containing 10% antibiotics (Penicillin-Streptomycin—Cat# 15140148, Gibco™, Waltham, MA USA) through the right ventricle of the heart. A fragment of the left lung was excised and immediately placed in the methacarn solution for fixation. After fixation, the lung fragments were dehydrated, clarified in xylol, embedded in paraffin, and sectioned. Thereafter, 5 µm-thick sections were prepared by slicing the tissue at 135 µm intervals. The sections were stained with hematoxylin and eosin (H&E), Grocott–Gomori methenamine silver (GMS), or mucicarmine (specific for *Cryptococcus* spp.) and evaluated using light microscopy (LMD6 system and motorized stage; Leica, Wetzlar, Germany; XYZ module LX; objectives HC PL FL 1.25×/0.04, 10×/0.30, 20×/0.40, 40×/0.60) and a DFC450 C digital camera (Leica). The images were processed using ImageJ Fiji open-source image processor software. Quantification of the number of yeast cells, average yeast area, yeast frequency, and compromised pulmonary parenchyma was performed in a semi-automated quantitative manner using ImageJ Fiji.

### 2.11. Evaluation of the Titanization Process of C. gattii in the Presence of Serum from Immunized Mice

*C. gattii* was cultivated for 24 h in SD medium at 30 °C. Thereafter, a *C. gattii* inoculum (concentration of 5 × 10^4^ cells/mL) was prepared using the titan cell medium (TCM) (5% Sabouraud with 5% fetal bovine serum (FBS) buffered with 50 mM morpholinepropanesulfonic acid (MOPS; pH 7.3) with 15 µM sodium azide), as reported by Trevijano-Contador [70]. The serum of untreated and treated mice was diluted (1:20) in TCM prior to incubation with *C. gattii* in a 96-well plate at 37 °C. *C. gattii* yeasts incubated in SD and TCM in the absence of serum were used as negative controls. The growth curve was determined by measuring absorbance at 540 nm after 6, 9, 12, 18, 21, and 24 h of incubation. After 24 h, the *C. gattii* culture was inactivated at 70 °C for 1 h, and the yeast size and cell concentration were measured by flow cytometry using the parameters forward scatter ((FSC)-HLin (median)) and cell/mL, respectively. The data were generated using three independent assays.

### 2.12. Statistical Analyses

Data were analyzed using Prism 9.0 (GraphPad Software). The normality of all statistical determinations was analyzed using the Shapiro–Wilk test. The homogeneity of variances was analyzed using the F-test for two groups or Bartlett’s test for three or more groups. The student’s *t*-test was applied to experiments with two groups, and analysis of variance (ANOVA) was applied to experiments with three or more groups when the samples had Gaussian distributions. For datasets with a non-normal distribution, the Mann–Whitney test was applied to experiments with two groups and the Kruskal–Wallis test for experiments with three or more groups. Differences between the means of groups were evaluated by one-way ANOVA followed by Tukey’s multiple comparisons test or Kruskal–Wallis test followed by Dunn’s multiple comparisons test. Differences were considered statistically significant at *p* < 0.05. Results are presented as mean ± standard deviation (SD) or median and interquartile range.

## 3. Results

### 3.1. Adjuvant Curdlan Slightly Improves the Switch of Immunoglobulin Isotypes in Immunized BALB/c Mice Challenged with C. gattii

Immunotherapeutic approaches using immunostimulatory agents or vaccines based on the cell wall structure of *Cryptococcus* spp. and/or its cytoplasmic components have shown positive effects against cryptococcosis [39,71,72]. An immunization strategy, which uses the specific antigens from *C. neoformans* and Dectin-1 ligand-coated particles, has shown potential for use against *C. neoformans* infection [73]. Therefore, we developed an immunization protocol using the selective Dectin-1 receptor agonist, curdlan, to stimulate innate immune cells and induce an efficient cellular and humoral immune response against *C. gattii*. BALB/c mice were divided into three groups: (i) untreated group (PBS only at all stages of the immunization protocol); (ii) immunized group (immunization with 2 × 10^7^ HK-*C. gattii*-thin yeast cells, i.n.); and (iii) curdlan group (immunostimulation with 200 µg/mouse i.p. of curdlan prior to immunization with 2 × 10^7^ HK-*C. gattii*-thin yeast cells, i.n.). On day zero, BALB/c mice were either immunostimulated with curdlan or not and, after three days, they were immunized with HK-*C. gattii*-thin. Additionally, two booster doses were administered at an interval of two weeks, on days 17 and 31 (Figure 1A). After 7 and 14 d.p.i., a blood sample was collected from each animal and used to measure the levels of serum IgM and IgG anti-GXM antibodies (Figure 1B,C). Mice immunized with HK-*C. gattii*-thin only showed a significant reduction in the levels of IgG anti-GXM at 14 d.p.i. compared to 7 d.p.i. (Figure 1C). However, there were no significant differences in the levels of IgM anti-GXM in any groups during the observation period (Figure 1B). Thereafter, the mice were inoculated (i.n.) with 1 × 10^5^
*C. gattii* yeast cells on day 45, and after two additional weeks (day 59), they were euthanized for analysis (Figure 1A).

On day 59, blood was collected from the mice to determine the serum levels of IgM and IgG anti-GXM antibodies. The levels of IgM and IgG anti-GXM did not differ among the groups (Figure 1D). For a more detailed analysis of the serum antibody profile, immunoglobulin isotyping was performed for IgM, IgG1, IgG2a, IgG2b, IgG3, and IgA, and the kappa and lambda light chains (Figure 2A–H). BALB/c mice immunostimulated with curdlan presented high levels of IgG1 and lambda light chains in comparison to the untreated group (Figure 2B). In contrast, IgA levels in the curdlan group were significantly lower than in the untreated and immunized groups (Figure 2F). The levels of IgM, IgG2a, IgG2b, and IgG3, and the kappa light chains, did not differ among the groups (Figure 2A,C–E,G). Taken together, the adjuvant curdlan promoted a slight change in the immunoglobulin isotype in BALB/c mice.

### 3.2. In Vitro Formation of Titan Cells Is Not Affected by Serum from BALB/c Mice That Received Adjuvant Curdlan

To evaluate whether the serum proteins from mice affected the growth and morphological transition of *C. gattii* yeasts to titan cells, we measured the optical density (OD) to generate a growth curve, and titan cell formation was determined using flow cytometry. The serum from mice in the different groups was incubated in vitro with *C. gattii*, and the absorbance at 540 nm after 6, 9, 12, 18, and 24 h of incubation (Figure 2I) was measured. The results demonstrated that incubation with serum from untreated, immunized, or curdlan groups did not decrease the growth of *C. gattii* (Figure 2I). Next, we cultured C. gattii in TCM and observed a marked reduction in its growth rate (Appendix A). However, we observed increased numbers of C. gattii cells with a large size (Appendix A) after cultivation in TCM compared to that after incubation in SD. This evaluation was standardized by flow cytometry (Figure 3A). The cell suspension was separated into three homogeneous populations, with distinct cell sizes considered within each gate and replicated for all groups analyzed (Appendix A). Our results showed that the serum obtained from untreated, immunized, or curdlan groups had no effect on the concentration of *C. gattii* cells grown in TCM (Figure 3B–D). Then, the formation of titan cells of *C. gattii* was not affected by the serum from mice treated with or without adjuvant curdlan.

### 3.3. Adjuvant Curdlan Altered the Cytokine Microenvironment in the Lungs of BALB/c Mice but Did Not Reduce the C. gattii Burden

To evaluate the efficacy of the immunization protocol, the animals were euthanized 14 days after *C. gattii* infection, and the lungs were collected to measure the CFU. There was no difference in the *C. gattii* burden between immunized mice that received or did not receive adjuvant curdlan, compared to untreated group (Figure 4A).

The binding of β-glucan to the Dectin-1 receptor promotes the innate immune response and results in the secretion of a range of cytokines [74]. It is important to highlight that the host immunity, both cell-mediated and humoral, is essential for managing *Cryptococcus* infections [75,76]. Therefore, the host’s immune response at the site of infection was evaluated after immunization with or without adjuvant curdlan. To measure the immune response, a lung fragment was homogenized, and the supernatant was used to measure the levels of pro- and anti-inflammatory cytokines, such as TNF-α, IFN-γ, IL-12p40, IL-6, IL-17, IL-10, and IL-4 (Figure 4B). The curdlan group showed an increase in the levels of TNF-α compared to the group that was only immunized (Figure 4B). Nevertheless, the immunized group showed a reduced level of TNF-α compared to that in the untreated group (Figure 4B and Appendix A). BALB/c mice that received curdlan showed reduced levels of IFN-γ compared to those from the untreated group (Figure 4B and Appendix A). The levels of IL-12p40, IL-6, IL-17, and anti-inflammatory cytokines did not differ significantly among the groups (Figure 4B and Appendix A). These findings demonstrate that the adjuvant curdlan prevented a decrease in the TNF-α levels induced after HK-*C. gattii*-thin administration but was not able to reduce the *C. gattii* burden in the lungs of mice.

### 3.4. Immunization with Heat-Killed C. gattii Increases the Cellularity of the Innate immunity Cells in the Lungs That Is Restored to Basal Levels by the Adjuvant Curdlan

Given that the immunization protocol was unable to reduce the pulmonary burden of *C. gattii*, we next evaluated the phenotype of the immune cells at the primary site of infection by flow cytometry (Figure 5A–L). A significant predominance of DCs, CD11b+ cells, and neutrophils was observed in the immunized-only group compared to the untreated group, based on the increased concentration of these cells in the lungs of both mice groups. However, the administration of curdlan restored the cellularity of DCs, CD11b+ cells, and neutrophils to basal levels in the lung tissue (Figure 5D–F,H, respectively). No changes in the frequency and concentration of CD3^+^ and F4/80^+^ cells were observed (Figure 5A,B,K–L).

### 3.5. Immunized BALB/c Mice That Received Adjuvant Curdlan Had a Reduction in the Relative Expression of Arginase-1 in the Lung after the C. gattii Challenge

The administration of curdlan in different experimental models has facilitated the differentiation of T cells towards the Th1 and Th17 profiles [56,57,58]. Additionally, we observed that using curdlan as an adjuvant significantly increased TNF-α levels in the lung tissue. TNF-α is a pro-inflammatory cytokine that facilitates the polarization of macrophages towards the M1 phenotype. Oliveira-Brito et al. had previously demonstrated that *Arginase*-1 expression is elevated in BALB/c mice after 14 days of infection with *C. gattii* [69]. Here, we used the same period of infection as that established by Oliveira-Brito et al. [69] and screened for the key transcription markers that are associated with innate and adaptive immunity cells (Figure 6A–F). Curdlan treatment promoted a reduction in the relative expression of *Arginase-1* when compared to the untreated group (Figure 6B). Moreover, there was no difference in the relative expressions of *iNOS*, *T-bet*, *GATA-3*, *ROR-γt*, and *FOXP3* among the groups (Figure 6A,C–F). Thus, the immunostimulation induced by curdlan reduced the relative expression of *Arginase-1* in the lung tissue of BALB/c mice that were challenged with *C. gattii*; however, it was unable to directly influence the differentiation of T helper cells towards the Th1 and/or Th17 profiles.

### 3.6. Curdlan Reduces the Average Area and Diameter of C. gattii Yeasts in the Lungs of BALB/c Mice

Given that an exacerbated pro-inflammatory response can result in tissue damage and that *C. gattii* has a predilection for the lung tissue, we performed the histopathological and morphometric analyses of mice lungs. The histopathological analysis revealed focal and multiple pulmonary nodular lesions in the sections of tissues from mice belonging to the untreated, immunized, and curdlan groups (Figure 7A–D). Specifically, the nodules were in the alveolar spaces and attached to the alveolar septa (Figure 7A,B). In all groups analyzed, we observed *C. gattii* yeasts and robust lymphocyte infiltration with the presence of neutrophils. Moreover, the *C. gattii* yeasts showed diffused distribution, which promoted their cytoplasmic vacuolization in macrophages. The formation of fungi-rich nodules was also observed in the lungs of mice from the untreated, immunized, and curdlan groups (Figure 7C–F; yellow arrows). The high frequency of *C. gattii* throughout the whole tissue was identified by staining with mucicarmin and Grocott techniques (Figure 7E,F). Figure 7 shows the panels that represent these findings for all groups studied.

The morphometric analysis revealed that the percentage of pulmonary parenchymal injury between mice from the curdlan and untreated groups did not differ significantly. Furthermore, mice that were immunized with HK-*C. gattii* only showed minor tissue damage compared to the animals from the untreated or curdlan groups (Figure 8A). During the histopathological analysis of the tissue, some groups appeared to have fewer yeast cells in the lung parenchyma compared to the others, even though yeast cells of different sizes were present. To further analyze this difference, we measured the number of yeast cells per area of the lung parenchyma, the average size of the yeast cells, and their relative frequency using the ImageJ Fiji processing package (Figure 8B–D). The curdlan group showed a higher number of yeast cells per area of the lung parenchyma compared to both the untreated and immunized only groups (Figure 8B). However, the yeast cells in the lungs of mice from the curdlan group had a lower average size and relative frequency than the yeast cells in the lung tissue of mice from the untreated group (Figure 8C,D). The same reduction was also observed in the case of tissues collected from the immunized-only group (Figure 8C,D). Although curdlan did not reduce the pulmonary burden of *C. gattii* in BALB/c mice, it reduced the diameter of *C. gattii* yeasts.

### 3.7. Immunized C57BL/6 Mice Treated with a Dectin-1 Ligand Had Higher Levels of IFN-γ and IL-10 in the Lung Tissue

It has been widely reported that BALB/c mice are more resistant to infection by *Cryptococcus* spp. than the C57BL/6 mouse strains [61,62,63,64]. To evaluate the effect of a Dectin-1 ligand in a more susceptible murine model, we treated C57BL/6 mice with curdlan or BGP before administering HK-*C. gattii*-thin and thereafter challenged them with *C. gattii*, as described in Section 2.3. After 14 days of infection, blood samples were collected from the mice to determine the levels of serum IgM and IgG anti-GXM antibodies (Figure 9). The IgM and IgG anti-GXM levels did not differ among the groups (Figure 9). In addition, lung fragments of mice treated with or without the Dectin-1 ligand were collected to measure the levels of pro- and anti-inflammatory cytokines. The curdlan treated group exhibited higher levels of IFN-γ and IL-10 than the immunized group (Figure 10B,E) and had increased levels of IL-12p40 and IL-10 compared to the untreated group (Figure 10C,E). The IL-4 levels in groups that were immunized with HK-*C. gattii*-thin combined or not with curdlan were lower than that in the untreated group (Figure 10F). The mice that received the adjuvant BGP had lower levels of IFN-γ than mice from the curdlan group (Figure 10B). Furthermore, the BGP group had higher levels of IL-12p40 and IL-10 than the untreated and HK-*C. gattii*-thin-only treated groups, respectively (Figure 10C,E). Finally, the pro-inflammatory cytokines induced by administration of the Dectin-1 ligands were associated with high levels of IL-10 in the lung tissue of C57BL/mice.

### 3.8. Adjuvant Curdlan Facilitates the Reduction in the C. gattii Burden in the Lungs of C57BL/6 Mice

Finally, the burden of *C. gattii* in the lungs of these mice was evaluated, and a significant reduction in the CFU was observed in the curdlan-treated group (Figure 11). These data suggest that immunostimulating C57BL/6 mice with curdlan promoted a balance between the pro- and anti-inflammatory responses, which effectively reduced the fungal burden in the lungs of these mice. On the other hand, there was no effect on the formation of titan-like cells after 18, 21, and 24 h of culture in TCM, in the presence of serum obtained from each mice group (Figure 12A). We also did not observe a difference in the concentration of *C. gattii* cells between the groups for the different sizes measured (Figure 12B–D).

## 4. Discussion

Innovative immunotherapeutic strategies using the cell surface and cytosolic components, which can act as antigens and induce humoral and cellular immune response against *Cryptococcus* spp., have demonstrated partial protection against cryptococcosis [69,70,71]. In contrast, particles coated with components from the fungal cell wall and loaded with specific antigens from the *Cryptococcus* spp. have shown promising effects [40,73,77,78]. The current study aimed to develop a vaccine, which uses the immunomodulatory activity of Dectin-1 agonists to support the cellular and humoral immune response induced after immunization with inactivated *C. gattii* yeasts. Therefore, we administered the aforementioned therapeutic to BALB/c and C57BL/6 mice and subsequently challenged them with *C. gattii*. Immunostimulation with curdlan increases serum IgG1 levels in BALB/c mice. Moreover, the adjuvant curdlan restored TNF-α levels and altered the frequency and concentration of CD11c+, Ly6G+, and CD11b+-positive cells in the lung tissue, which may be associated with a reduction in the relative expression of *Arginase-1* in BALB/c mice. Interestingly, these effects could not reduce the *C. gattii* burden in the lungs of BALB/c mice. Nevertheless, BALB/c mice from the curdlan group showed a reduction in the average area and diameter of yeast cells in the lung tissue, thereby reducing the titanization process of *C. gattii*. In contrast, C57BL/6 mice, which are more susceptible to *Cryptococcus* spp. infection, showed a reduction in the pulmonary burden of *C. gattii* when they received curdlan before immunization. The adjuvant curdlan favored an increase in the levels of IFN-γ and IL-10, as well as resulted in the higher relative expression of *ROR*-γt in the lungs of C57BL/6 mice. This work provides proof of concept for using the Dectin-1 ligand in the immunization strategy against cryptococcosis. As evidenced by the distinct murine lineages studied, our study highlights the importance of a balanced immune response to control *C. gattii* infection.

*C. gattii* infection affects both immunosuppressed and healthy individuals [7]. Therefore, it must draw the attention of public health policies, given the history of the *C. gattii* infection outbreaks that have already occurred in underdeveloped and developed countries [79,80,81,82,83]. This problem is further exacerbated by the difficulty in developing antifungals because agents that are toxic to the fungi also trigger side effects in patients [84,85]. Therefore, it is crucial to develop immunotherapeutic strategies that stimulate the host immune system in response to fungal infections. Fungal pathogens have also developed efficient mechanisms to escape the host’s immune response by altering the shape, size, composition, and structure of their cell wall or masking their PAMPs to impair recognition by cells of the immune system [86]. Immunostimulation with curdlan or BGP can enhance the recognition of antigens by the immune system via specific receptors, such as Dectin-1. Thus, the present study aimed to develop an immunotherapeutic strategy, which uses selective Dectin-1 agonists to target the receptor and promote the induction of protective immunity against the pathogen. Studies evaluating prophylactic strategies against different fungi, such as *Paracoccidioides brasiliensis* and *Candida albicans*, have reported a reduction in the CFU and an increase in phagocytosis and killing in mice treated with immunomodulators three days before infection [42,43,44,87]. Priming DCs with the P10 peptide and then administering these cells to mice (one day before infection) also caused a significant reduction in the pulmonary fungal burden [46]. Considering that prior administration of such immunomodulators has shown positive effects against different fungal infections, the current study evaluated the potential for using a combination of curdlan or BGP with heat-killed *C. gattii* as a vaccination strategy. To improve the recognition of *C. gattii* during the immunization process, we immunized mice with HK-*C. gattii* with a thin capsule. This minimized the immunogenic and immunomodulatory effects of GXM and other polysaccharides in the capsules of *Cryptococcus* spp. Immunization with HK-*C. gattii* yeast can enable specific cellular and humoral immune responses.

After 14 days of infection, BALB/c mice immunostimulated with curdlan showed elevated serum IgG1 levels compared to untreated mice. Murine IgG1 can bind to the GXM of *Cryptococcus* [88], thereby allowing the engulfment of larger yeast cells. Although the size of the macrophages is compatible with the typical diameter of *Cryptococcus* yeast cells (from 4 to 10 µm), the conditions in the host microenvironment can induce yeast growth through the titanization process. This results in yeast cells with a diameter greater than 10 µm [89,90,91], requiring opsonization to engulf such particles that are larger than phagocytic cells [92,93].

A reduction in the serum IgA levels was also observed in the curdlan group and was associated with a possible influx of IgA into the lung tissue. Trevijano-Contador et al. have demonstrated that IgM and IgA can bind to the components in the *Cryptococcus* spp. cell wall, resulting in reduced yeast titanization [70]. Next, we performed in vitro studies to evaluate the lytic effects of the plasma proteins in the serum of immunized mice on the growth of *C. gattii*. We observed no changes in cell growth or concentration. In contrast, BALB/c mice immunostimulated with curdlan showed a reduction in the average area and diameter of yeast cells, which may lead to a better prognosis of the disease and favor recognition by phagocytes [94].

Curdlan immunostimulation restored TNF-α levels in the lungs of BALB/c mice and reduced the relative expression of *Arginase-1*; however, iNOS expression remained unchanged. These findings could contribute to our understanding of the control of *C. gattii* infection and demonstrate that curdlan administration induces a balance between pro- and anti-inflammatory responses. An imbalance in the host immune response can increase the number of yeast cells per pulmonary area. Voelz et al. demonstrated that the hyperproliferative *C. gattii* genotypes, such as the genotype adopted in the present study, have intracellular proliferation rates dependent on the levels of the reactive oxygen species (ROS) in macrophages [95]. Inhibiting the pro-inflammatory response in macrophages blocked the increase in intracellular proliferation of hyperproliferative genotype strain [95]. Additional studies will be required to determine the mechanism by which the rate of proliferation of the pathogen may be associated with the presence of ROS. Nevertheless, it has been suggested that phagosome modulation and changes in the morphology and gene expression of the pathogen’s mitochondria [75,95,96] may explain this phenomenon. This information corroborates the data in the present study, where an imbalance between the pro- and anti-inflammatory responses could be linked to the increased size of the yeast cells in the lungs and a subsequent increase in the injury level of the lung parenchyma.

A recent study reported the ability of curdlan to induce neutrophil recruitment [97]. In contrast to this study, we observed no increase in the frequency of neutrophils in the lung tissue of curdlan-treated mice. In vitro studies have demonstrated the role of neutrophils in cryptococcosis and suggest that neutrophils may kill yeast cells through oxidative and non-oxidative mechanisms [98,99]. However, Mednick et al. demonstrated a contradictory contribution of neutrophils during *Cryptococcus* infection in vivo, whereby the depletion of neutrophils in BALB/c mice challenged with a lethal inoculum increased the survival of these mice, possibly through the modulation of adaptive immune response [100]. Thus, augmented neutrophil influx into the lungs may not be required for the curdlan-dependent control of the pulmonary burden of *C. gattii*.

To obtain insight into types of immune responses induced after immunization with the adjuvant Dectin-1 ligand, we included C57BL/6 mice, which are susceptible to *Cryptococcus* spp. infection [63], in the current study. C57BL/6 mice infected with *C. neoformans* showed ineffective pulmonary clearance and Th2 responses [63]. Interestingly, C57BL/6 mice that received curdlan prior to immunization had increased levels of IL-12p40, IFN-γ, and IL-10 (Figure 10), as well as higher expression of *ROR*-γt (Appendix A). Collectively, these changes favored a reduction in the pulmonary *C. gattii* burden in the curdlan group compared to that in untreated mice (Figure 11). Conversely, the adjuvant BGP did not improve the control of *C. gattii* infection (Figure 11). Milam et al. demonstrated that treating *C. neoformans*-infected C57BL/6 mice with TNF-α induced a Th1 response and facilitated the M1 polarization of macrophages, which resulted in the control of fungal burden in the early stages of infection [101]. Our data showed that immunizing C57BL/6 mice with the adjuvant curdlan favored Th1 and Th17 cell differentiation and decreased the levels of IL-4. Previous studies have shown that C57BL/6 mice infected with *Cryptococcus* sp. develop a Th1 response that promotes the clearance of infection. Importantly, the absence of IL-4 further improves this response [102].

The current work demonstrates that host-specific differences in the immune response affect the efficacy of the immunization strategy described in our study, as verified in BALB/c and C57BL/6 mice. The successful control of fungal infection in C57BL/6 mice may be explained by the induction of a balanced immune response in this mouse strain. Our data showed that adherent lung cells in BALB/c mice are more pro-inflammatory than in C57BL/6 mice (Appendix A). It is likely that the pronounced pro-inflammatory state of the innate immune cells in the lungs of BALB/c mice did not facilitate the induction of protective immunity in response to the immunization strategy described in the current study. Our findings align with the study by Chen et al. [63], which highlights the differences in the immune response to cryptococcal pulmonary infection in the case of the BALB/c and C57BL/6 mouse strains. The balanced immune response induced in C57BL/6 mice favors the successful immunization protocol associated with adjuvant curdlan to treat *C. gattii* infection. Finally, these findings will have implications for developing new immunotherapeutic approaches as they suggest that evoking an exacerbated immune response may be detrimental when treating *C. gattii* infection.

## Figures and Tables

**Figure 1 vaccines-10-00620-f001:**
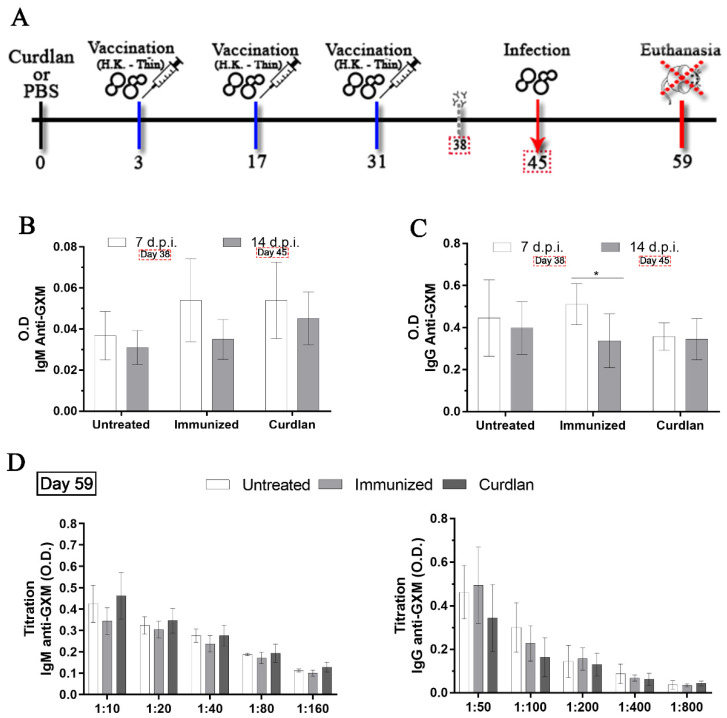
Evaluation of the immunotherapeutic strategy combined with the adjuvant curdlan against *C. gattii* challenge. (**A**) BALB/c mice were intraperitoneally (i.p.) administered curdlan (200 μg/mouse) or phosphate-buffered saline (PBS) on day 0. On day 3, they were immunized with 2 × 10^7^ yeast cells of heat-inactivated *Cryptococcus gattii* (HK-*C. gattii*-thin) or administered PBS intranasally (i.n.), and booster doses were administered after 14 days. (**B**,**C**) On days 38 and 45 (7 and 14 days post immunization (d.p.i.), respectively), the levels of serum immunoglobulin M (IgM) and IgG anti-glucuronoxylomannan (GXM) antibodies were measured (dilutions of 1:200). On day 45, the mice were i.n.-infected with 1 × 10^5^
*C. gattii* yeast cells. (**D**) On day 59 (14 days post infection), the mice were euthanized and their serum titrated to measure the IgM and IgG anti-GXM levels (IgM dilutions of 1:10, 1:20, 1:40, 1:80, and 1:160; and IgG anti-GXM dilutions of 1:50, 1:100, 1:200, 1:400, and 1:800). The results are expressed as mean ± SD. * *p* < 0.05.

**Figure 2 vaccines-10-00620-f002:**
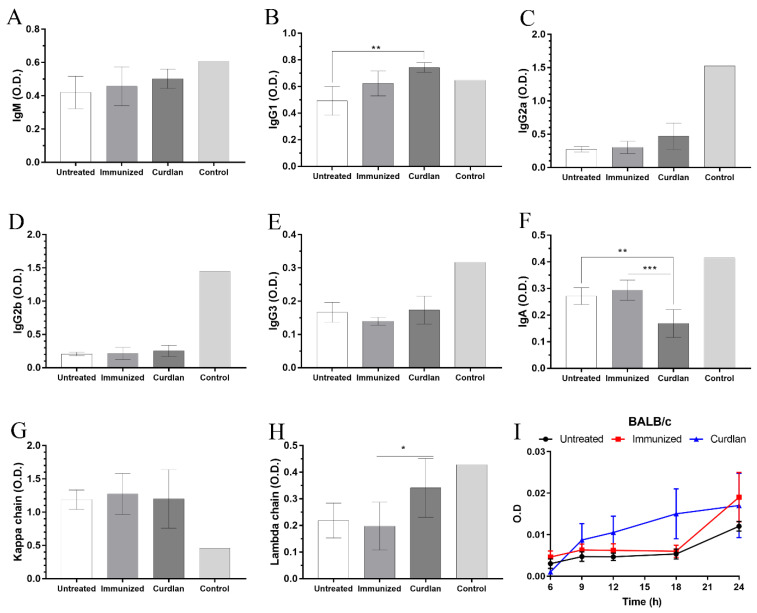
Isotyping of serum immunoglobulins in BALB/c mice and the effect of serum on the growth curve of *C. gattii* cultivated in titan cell medium (TCM). BALB/c mice immunostimulated with curdlan and/or immunized with HK-*C. gattii*-thin were euthanized 14 days post infection with *C. gattii*. The serum was collected and used for both isotyping the immunoglobulins and incubating with *C. gattii* to determine its growth curve. The (**A**) IgG1, (**B**) IgG2a, (**C**) IgG2b, (**D**) IgG3, (**E**) IgM, (**F**) IgA isotypes, and the (**G**) kappa and (**H**) lambda light chains were evaluated. (**I**) *C. gattii* R265 was cultivated at 37 °C for 24 h in TCM, and the growth curves were determined in the presence of serum from mice belonging to the untreated (black circle), immunized (red square), and curdlan groups (blue triangle). The results are expressed as mean ± SD. * *p* < 0.05, ** *p* < 0.01, and *** *p* < 0.001.

**Figure 3 vaccines-10-00620-f003:**
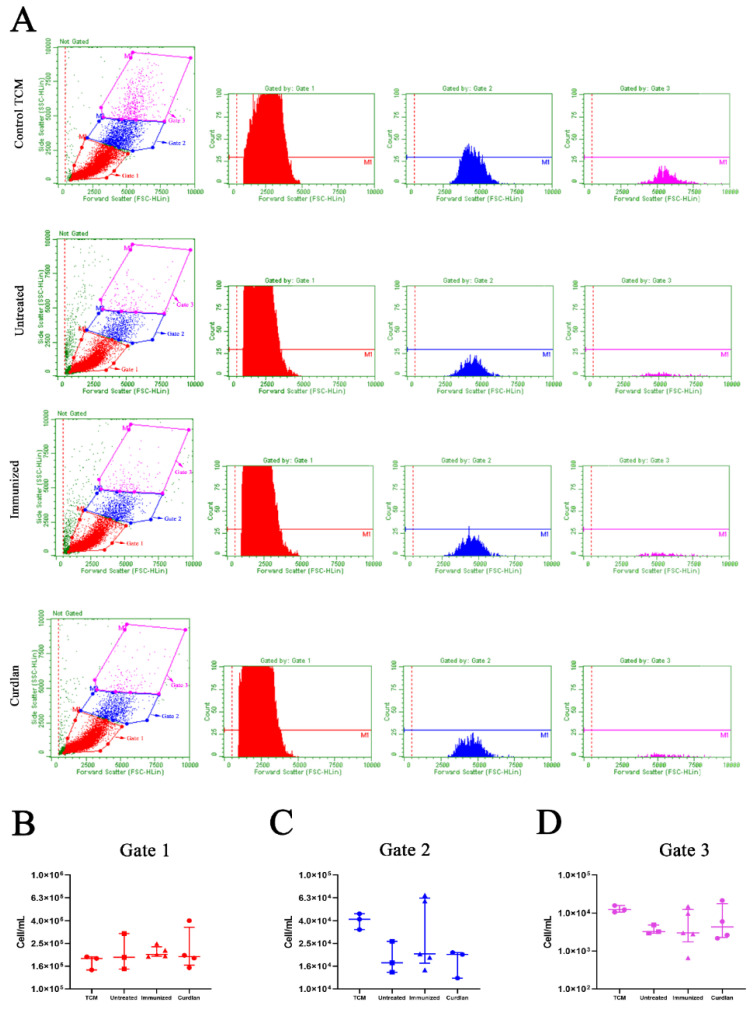
Evaluation of the size and concentration of *C. gattii* cultured in TCM in the presence of serum from BALB/c mice treated with the adjuvant curdlan. The BALB/c mice received curdlan or PBS and were immunized with HK-*C. gattii*-thin, before being challenged with *C. gattii*. After 14 days of infection, the serum was collected and incubated with *C. gattii* cultured in TCM. After 24 h of incubation, the *C. gattii* cells were analyzed by flow cytometry. The size and concentration of the three populations of *C. gattii* cells, as delimited by gates 1 (red; (**A**,**B**)), 2 (blue; (**A**,**C**)), and 3 (pink; (**A**,**D**)), were measured. The FSC-HLin was the parameter used to normalize the size among the groups to express the cell concentration (cell/mL). Results are expressed as medians with interquartile ranges.

**Figure 4 vaccines-10-00620-f004:**
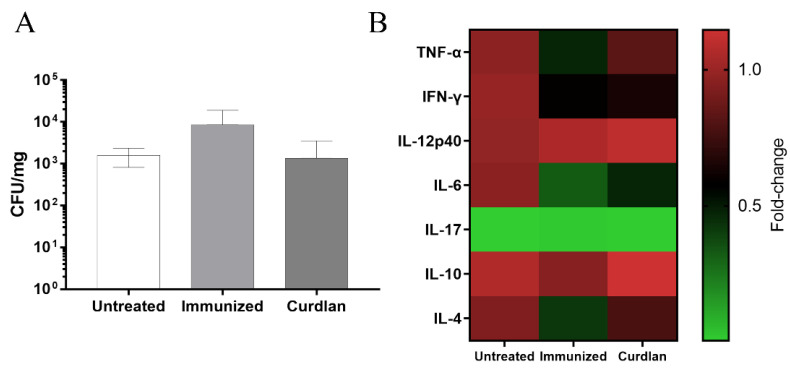
Measurement of the pulmonary fungal burden and cytokine levels in the lungs of immunized BALB/c mice after *C. gattii* challenge. After administering curdlan or PBS, the mice were immunized with HK-*C. gattii*-thin. On day 45 (14 d.p.i.), the mice were challenged with *C. gattii*. On day 59 (14 d.p.i.), the lungs of the mice were harvested and homogenized. The supernatants were used to (**A**) determine the colony-forming units (CFU), normalized to the lung mass (CFU/mg), and (**B**) measure the levels of cytokines tumor necrosis factor-α (TNF-α), interferon-γ (IFN-γ), interleukin (IL)-12p40, IL-6, IL-17, IL-10, and IL-4 using enzyme-linked immunosorbent assay (ELISA). Values are expressed as mean ± SD or as medians with maximum and minimum values.

**Figure 5 vaccines-10-00620-f005:**
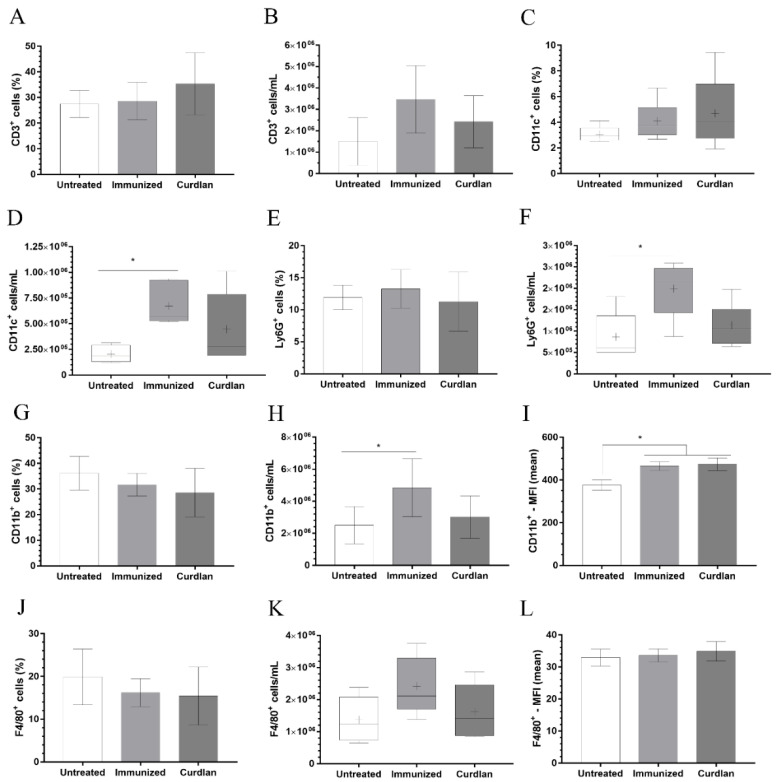
Phenotyping of the leukocytes in the lungs of mice treated with curdlan and immunized with HK-*C. gattii*. After administration of curdlan or PBS, the mice were immunized with HK-*C. gattii*-thin and infected with viable *C. gattii*. The lungs of the infected mice were harvested at 14 days post infection. Pulmonary leukocytes were obtained by dissociating the tissue with collagenase. The cell suspensions were used to determine the frequency (%) and concentration (cell/mL) of T lymphocytes (**A**,**B**), CD11c^+^ (**C**,**D**), Ly6G^+^ (**E**,**F**), CD11b^+^ (**G**,**H**), and F4/80^+^ (**J**,**K**). The mean fluorescence intensity (MFI) of (**I**) CD11b^+^ and (**L**) F4/80^+^ cells was recorded using flow cytometry. The relative and absolute frequencies of these cells were determined in the lung samples. The results are expressed as both, in percentage and cells/mL, and are represented as mean ± SD or as median with maximum and minimum values. Comparisons were made between groups for the same period of infection. * *p* < 0.05.

**Figure 6 vaccines-10-00620-f006:**
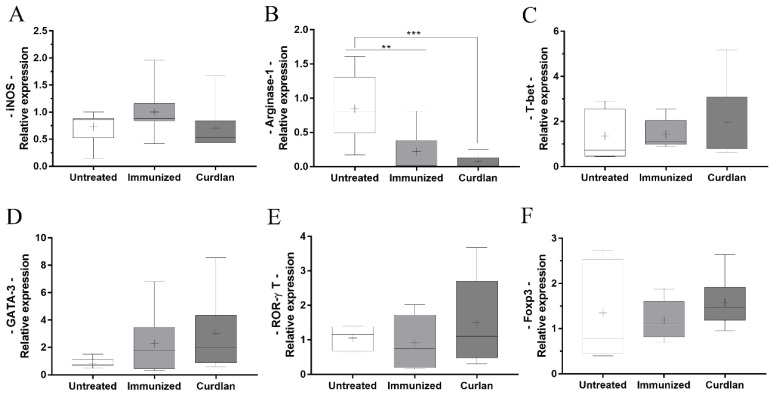
Relative expression of transcription factors related to macrophage polarization and T-helper cell differentiation in the lungs of BALB/c mice treated with curdlan and immunized with HK-*C. gattii*. Lungs were harvested from the mice 14 d.p.i., and total RNA was extracted from the lungs. The total RNA was reverse-transcribed into complementary DNA (cDNA), and the relative expression levels of (**A**) *iNOS*, (**B**) *Arginase-1*, (**C**) *T-bet*, (**D**) *GATA-3*, (**E**) *ROR-γt*, and (**F**) *FOXP3* were determined by real-time quantitative polymerase chain reaction (qRT-PCR). The values were normalized to *β-actin* expression. The results are expressed as median, maximum, and minimum values. Comparisons were made between groups for the same period of infection. ** *p* < 0.01, and *** *p* < 0.001.

**Figure 7 vaccines-10-00620-f007:**
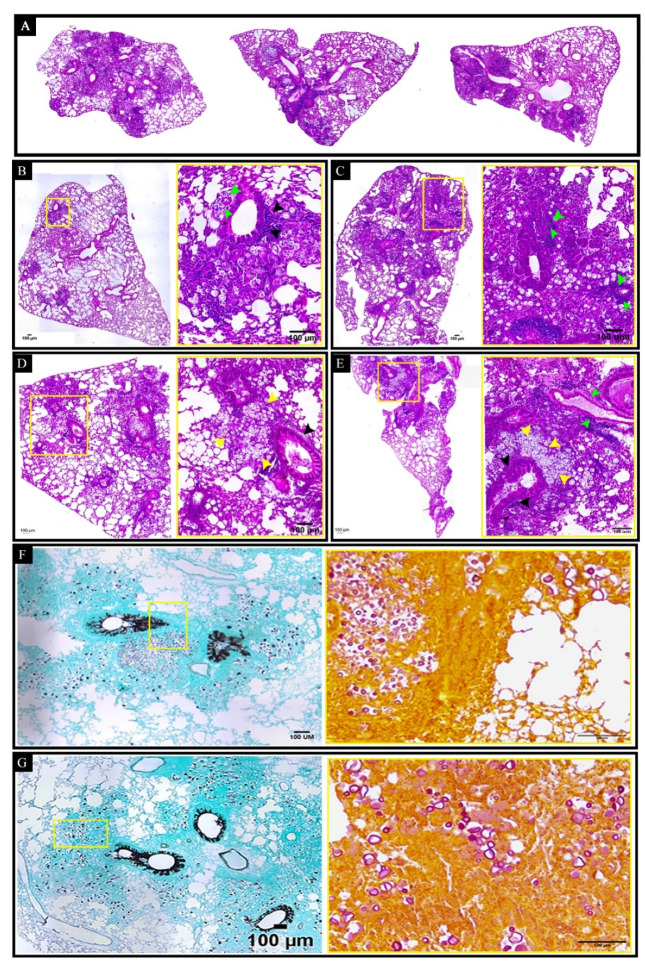
Histopathological analysis of the lungs of BALB/c mice treated with curdlan and immunized with HK-*C. gattii*. (**A**) The panels show representative lung sections from untreated mice (left side) or those that were immunized with HK-*C. gattii* in the presence (right size) or absence (middle size) of curdlan. (**B**–**E**) The total lung (10× magnification) and lung sections (40× magnification) have been shown for each panel. The sections were stained with hematoxylin and eosin (H&E). Perivasculitis (green arrows) and peribrochiolitis (black arrows) were observed, with exudates primarily composed of mononuclear leukocytes. Inflammatory infiltrate composed of mononuclear leukocytes and neutrophils filling the alveolar space contained oval to rounded (yellow arrows) yeast cells diffusely distributed or restricted to the vacuolated cytoplasm of macrophages. The nodules consisted of high concentrations of aggregated *C. gattii* yeasts (yellow arrows). (**F**,**G**) The panels show representative lung sections for the different groups. The sections were stained with Grocott (left side, 10× magnification) and Mucicarmin (right side; 20× magnification) stains to visualize the distribution of fungi. The images presented are representative fields of the different groups. Five mice were analyzed per group. Magnification bar: 100 µm.

**Figure 8 vaccines-10-00620-f008:**
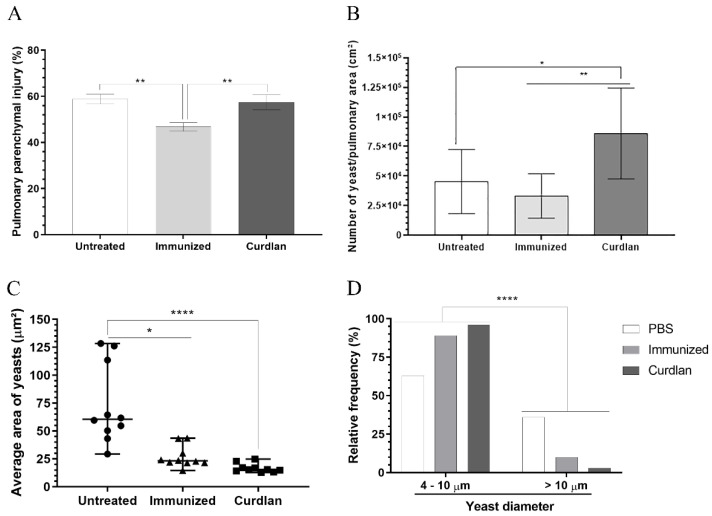
Morphometric analysis of the lung tissue in BALB/c mice treated with curdlan, immunized with HK-*C. gattii*, and infected with viable *C. gattii*. Lungs were perfused with PBS and harvested from the BALB/c mice 14 days post infection. Lung fragments were fixed with methacarn, dehydrated in an alcohol battery, diaphanized with xylol, and embedded in paraffin. Serial 5 µm cuts were made and stained with Grocott-Gomori methenamine silver (GMS). (**A**) Pulmonary parenchyma injury (%), (**B**) quantification of the number of yeast cells/pulmonary area (cm^2^), (**C**) average yeast cell area (µm^2^), and (**D**) yeast relative frequency were evaluated in a semi-automated quantitative way using ImageJ Fiji. The data were analyzed using the X^2^ test (Goodness-of-fit test). Values were expressed as mean ± SD. * *p* < 0.05, ** *p* < 0.01, and **** *p* < 0.0001.

**Figure 9 vaccines-10-00620-f009:**
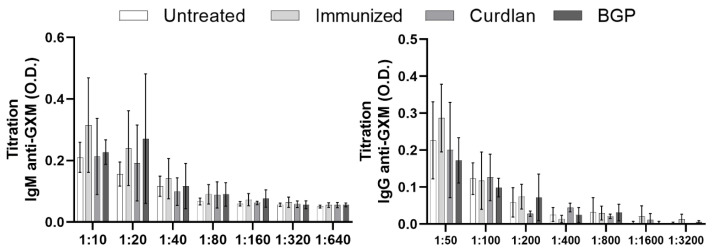
The levels of IgM and IgG anti-GXM antibodies in the serum and growth curve for *C. gattii* after incubation with serum from immunized C57BL/6 mice. After administering Dectin-1 ligands or PBS, C57BL/6 mice were immunized with HK-*C. gattii*-thin. On day 45 (14 d.p.i.), the mice were challenged with *C. gattii*. On day 59 (14 d.p.i.), mice serum and lungs were collected. Several dilutions of the serum were used to measure the levels of IgM and IgG anti-GXM antibodies (IgM = 1:10, 1:20, 1:40, 1:80, 1:160, 1:320, and 1:640; IgG = 1:50, 1:100, 1:200, 1:400, 1:800, 1:1600, and 1:3200).

**Figure 10 vaccines-10-00620-f010:**
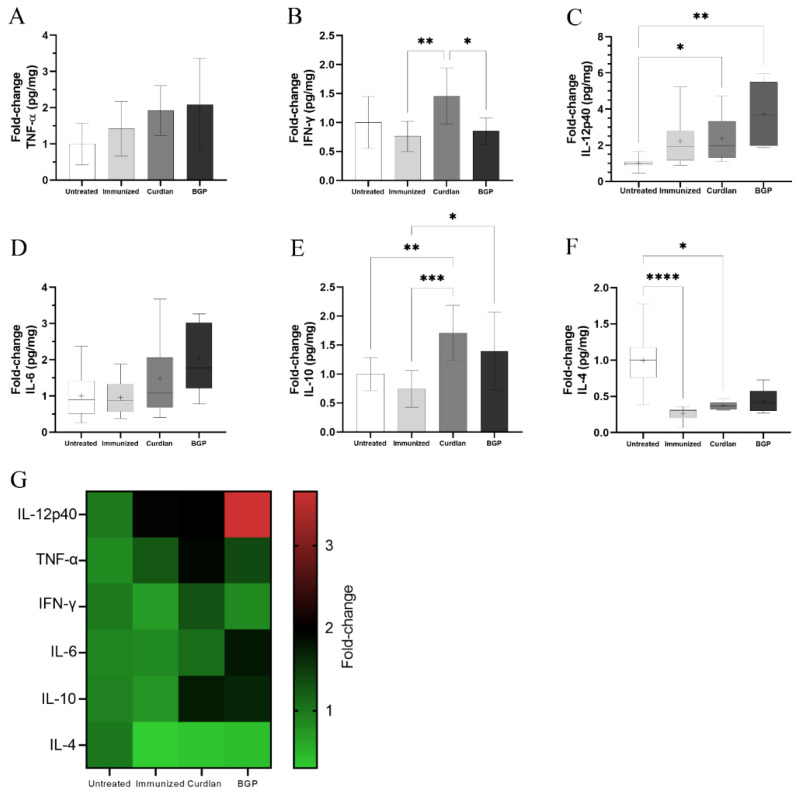
Measurement of cytokine levels in the lungs of immunized C57BL/6 mice challenged with *C. gattii*. After the administration of curdlan, β-glucan peptide (BGP), or PBS, C57BL/6 mice were immunized with HK-*C. gattii*-thin. On day 45 (14 d.p.i.), the mice were challenged with *C. gattii*. On day 59 (14 days post infection), mice lungs were harvested and homogenized, and the supernatants were used for measuring the levels of (**A**) TNF-α, (**B**) IFN-γ, (**C**) IL-12p40, (**D**) IL-6, (**E**) IL-10, and (**F**) IL-4 cytokines via ELISA. (**G**) Fold change (1–3 times) in the cytokine levels in all groups (in comparison to the untreated group) are represented in the heat map. Results are expressed as mean ± SD or as medians with interquartile ranges. * *p* < 0.05, ** *p* < 0.01, *** *p* < 0.001, and **** *p* < 0.0001.

**Figure 11 vaccines-10-00620-f011:**
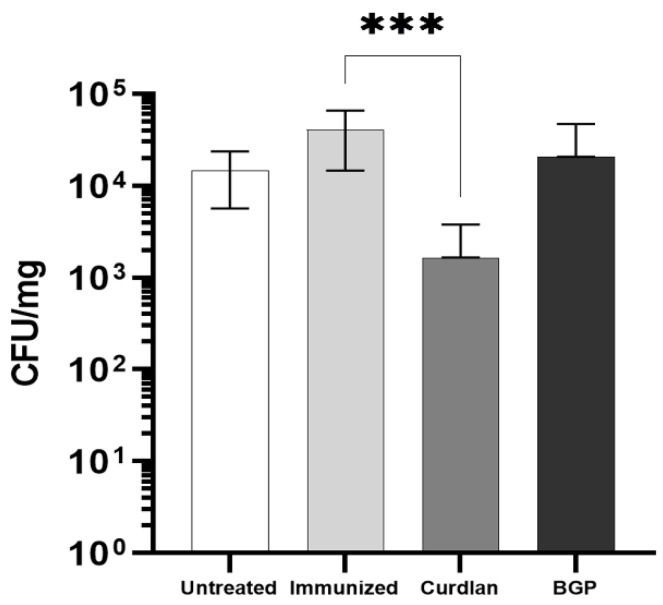
Quantification of the pulmonary fungal burden in C57BL/6 mice subjected to the HK-*C. gattii* immunization protocol, with or without the administration of curdlan. C57BL/6 mice received curdlan (200 μg/mouse), BGP (200 μg/mouse), or PBS i.p. on day 0. On days 3, 17, and 31, they were immunized with 2 × 10^7^ HK-*C. gattii*-thin yeast cells or administered PBS i.n., with an interval of 14 days between immunizations. On day 45 (14 d.p.i.), the mice were infected with *C. gattii*. On day 59 (14 days post infection), the mice were euthanized, and the pulmonary fungal burden was quantified using the CFU (colony-forming units) assay and normalized to the lung mass (CFU/mg). The results are expressed as mean ± SD. *** *p* < 0.001.

**Figure 12 vaccines-10-00620-f012:**
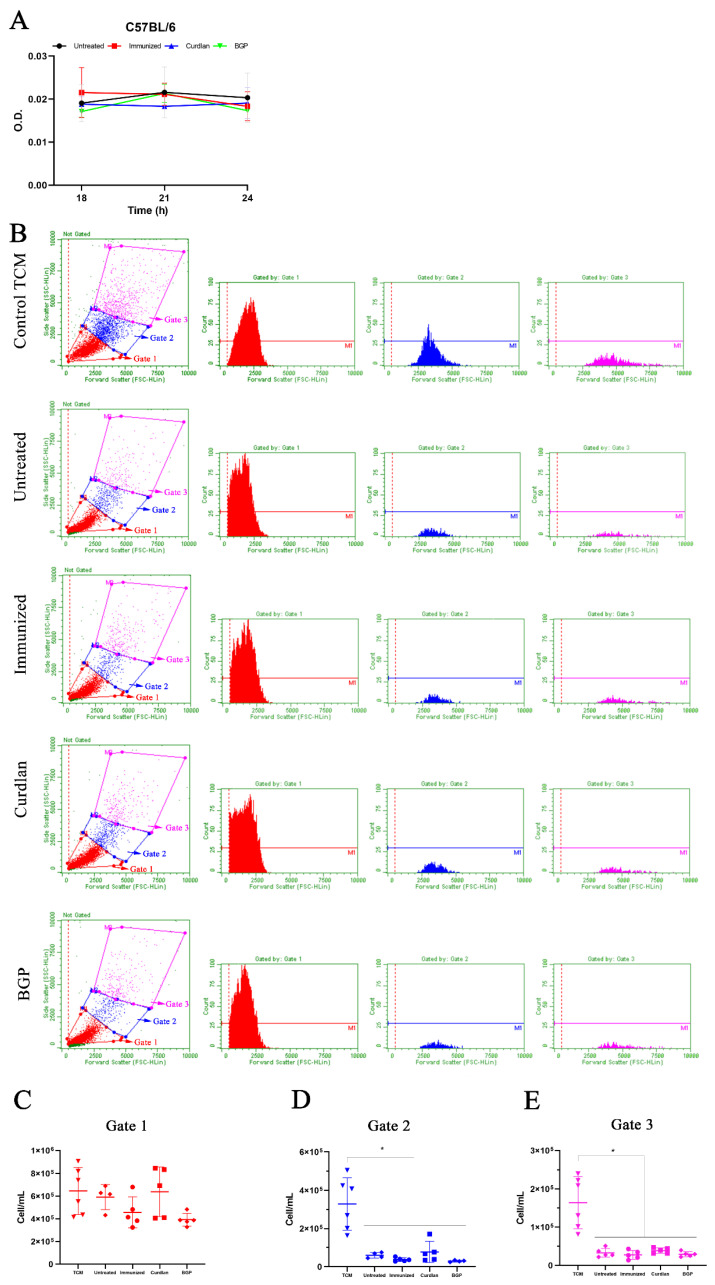
Evaluation of the growth and concentration of *C. gattii* cells cultured in TCM and incubated with serum from C57BL/6 mice treated with the Dectin-1 ligand. C57BL/6 mice that had received curdlan, BGP, or PBS were immunized with HK-*C. gattii*-thin before being challenged with *C. gattii*. Serum was collected after 14 days of infection and incubated with *C. gattii* in TCM. (**A**) *C. gattii* R265 was cultured at 37 °C for 18, 21, and 24 h in TCM, and the growth curves were determined in the presence of serum from untreated (black circle), immunized (red square), curdlan (blue triangle), and BGP groups (green triangle). (**B**–**E**) After 24 h of incubation, *C. gattii* was analyzed by flow cytometry. The cell size and concentration of the three populations as defined by gates 1 (red; (**B**,**C**)), 2 (blue; (**B**,**D**)), and 3 (pink; (**B**,**E**)) were measured. Cell concentration is expressed in cell/mL. Results are expressed as mean ± SD, or as medians, and maximum and minimum values. * *p* < 0.05.

## Data Availability

Not applicable.

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
