# Peer review of "Adjuvant Curdlan Contributes to Immunization against Cryptococcus gattii Infection in a Mouse Strain-Specific Manner"

_vaccines, 2022, doi:10.3390/vaccines10040620_

Round 1

Reviewer 1 Report

The Ms is extremely interesting and brings new information about the important topic of study.
I would particularly ask the authors to add in the introduction some information about Cryptococcus, emphasizing its polysaccharide capsule, which has important immunomodulatory mechanisms. this can be done using 3 lines.

In recent years, a low action of the main antifungicides used has been observed and, therefore, it is important to study for the development of new therapeutic techniques for this type of infection.
This month the authors developed an interesting immunization strategy. The results presented here elegantly provide important contributions to the advancement of control of cryptococcosis caused by Cryptococcus gattii. MS is extremely interesting. Thus, it is ready for publication in Vaccines magazine

I also leave below some references that can be used:

1) Immunomodulatory Role of Capsular Polysaccharides Constituents of Cryptococcus neoformans. Decote-Ricardo D, LaRocque-de-Freitas IF, Rocha JDB, Nascimento DO, Nunes MP, Morrot A, Freire-de-Lima L, Previato JO, Mendonça-Previato L, Freire-de-Lima CG. Front Med (Lausanne). 2019 Jun 19;6:129. doi: 10.3389/fmed.2019.00129.

Author Response

Please, find attached the responses.

Thanks

Reviewer 2 Report

Oliveira-Brito et al. aim to develop a vaccine, which uses the immunomodulatory activity of Dectin-1 agonists to support the cellular and humoral immune response induced after  immunization with heat-killed Cryptococcus gattii yeasts. The immunization strategy  in this manuscript uses two Dectin-1 agonist compounds, either, a linear beta-glucan curdlan derived from the bacterium Alcaligenes, or β-glucan peptide (BGP), a polysaccharide with a highly branched glucan portion derived from the basidiomycete Trametes. The ability of curdlan or BGP to serve as an immunotherapeutic agent against C. gattii has not been investigated in the literature. In the study described in Oliveira-Brito et al., BALB/c or 18 C57BL/6 mice received curdlan or BGP before immunization with inactivated C. gattii, and the mice were then infected with viable C. gattii on day 14 post-immunization and euthanized 14 days following infection. Interestingly, the two strains of mice displayed different reactions to the immunization. In BALB/c mice, the relative frequency and average area of C. gattii titan cells in the lungs of curdlan-treated mice were reduced; however, the pulmonary fungal burden was not statistically reduced. In the C57BL mice,  the adjuvant curdlan reduced the pulmonary C. gattii burden. Regardless of the mice strain, addition of curdlan as an adjuvant played a role in controlling the C. gattii infection.

Specific Comment:

In Figure 8D, there is a statistical difference in fungi with 4-10 micron diameter versus fungi with a greater than 10 micron diameter regardless of whether the BALB/c mice are treated with PBS, curdlan, or immunized with inactivated C. gattii. Is there a statistical difference in the relative frequency of fungi with greater than 10 micron diameter between the three treatments?

Author Response

(The authors gave the same response as above.)
